# Breast Cancer Molecular Subtyping in Practice: A Real-World Study of the APIS Breast Cancer Subtyping Assay in a Consecutive Series of Breast Core Biopsies

**DOI:** 10.3390/ijms25052616

**Published:** 2024-02-23

**Authors:** Silvana Di Palma, Panagiotis Koliou, Alex Simonovic, Daniela Costa, Catherine Faulkes, Brenda Kobutungi, Felicity Paterson, Jonathan David Horsnell, Farrokh Pakzad, Tracey Irvine, Polly Partlett, Elizabeth Clayton, Nadine Collins

**Affiliations:** 1Department of Cellular Pathology, Berkshire & Surrey Pathology Services, The Royal Surrey Hospital NHS Foundation Trust, University of Surrey, Egerton Road, Guildford GU2 7XX, UK; alexandra.simonovic@nhs.net; 2Department of Oncology, The Royal Surrey Hospital NHS Foundation Trust, Egerton Road, Guildford GU2 7XX, UK; panagiotis.koliou@nhs.net (P.K.); felicity.paterson1@nhs.net (F.P.); 3Molecular Diagnostics, Berkshire & Surrey Pathology Services, The Royal Surrey Hospital NHS Foundation Trust, Egerton Road, Guildford GU2 7XX, UK; daniela.costa1@nhs.net (D.C.); c.faulkes@nhs.net (C.F.); b.kobutungi@nhs.net (B.K.); nadine.collins@nhs.net (N.C.); 4Breast Unit, The Royal Surrey Hospital NHS Foundation Trust, Egerton Road, Guildford GU2 7XX, UK; j.horsnell@nhs.net (J.D.H.); f.pakzad@nhs.net (F.P.); traceyirvine1@nhs.net (T.I.); ppartlett@nhs.net (P.P.); elizabethclayton1@nhs.net (E.C.)

**Keywords:** breast cancer, immunohistochemistry, ER, PR, HER-2, Ki-67, APIS mRNA molecular assessment

## Abstract

The APIS Breast Cancer Subtyping Kit is an mRNA-based assessment of the seven parameters including three biomarkers routinely assessed in all the newly diagnosed breast cancers (BC), oestrogen receptor (ER), progesterone receptor (PR) and HER-2 and an additional four genes that create a novel proliferation signature, MKI67, PCNA, CCNA2 and KIF23. Taken together, the data are used to produce a molecular subtype for every sample. The kit was evaluated against the current standard protocol of immunohistochemistry (IHC) and/or in situ hybridisation (ISH) in breast cancer patients. The data were presented at the weekly breast multidisciplinary team (MDT) meeting. A total of 98 consecutive cases of pre-operative breast cancer core biopsies and two core biopsies of nodal metastases yielding 100 cases were assessed. IHC and APIS results were available for 100 and 99 cases. ER was concordant in 97% cases, PR was concordant in 89% and HER-2 results were concordant with IHC/ISH in 100% of the cases. Ki-67 IHC was discordant in 3% of cases when compared with MK167 alone but discordant in 24% when compared with the four-gene proliferation signature. In conclusion, our study indicates that the APIS Breast Cancer Subtyping Kit is highly concordant when compared to the results produced for ER/PR/HER-2 by IHC and/or ISH. The assay could play a role in the routine assessment of newly diagnosed breast cancer (BC) specimens.

## 1. Introduction

Currently, the use of IHC and/or ISH is the standard of care for the routine assessment of biomarkers in breast cancers. This assessment allows subdivision of BC into several groupings: hormone receptor (ER/PR) positive/negative, HER-2-positive equivocal and negative and triple negative breast cancers (TNBC).

Studies by Perou et. al. [1] reported that BC can also be classified into several molecular subgroups based on gene expression profiles: luminal, HER-2, basal-like and normal breast-like cancers. Subsequently, these groups have been shown to have distinct clinical behaviours and responses to chemotherapy [2]. As most histology laboratories have not had direct access to molecular testing, IHC tests have been used as a surrogate for gene expression analysis to reproduce these gene expression profiles [3]. These surrogate IHC tests have also been applied to carcinomas of salivary gland origin [4].

The search for more refined tests in breast cancer applies in all fields from non-invasive diagnostic approaches [5] as well as invasive procedures such as core biopsy with a focus on proliferation [6] or aberration of cell cycle [7].

However, it is accepted that BC gene expression profiles are more complex than this and this has led to the development of RNA expression profile tests such as OncotypeDX, Endopredict and Prosigna (based on PAM50). These tests are all now included in the NHSE Cancer Test Directory (NHS England Website) on surgically resected BC tissue in a particular cohort of patients. They utilise the expression data from between 12 and 50 genes, depending on which profiling test is being run. Unsurprisingly, there are significant discordances between subtyping with these molecular tests and IHC-based testing [8].

In an attempt to further refine IHC subtyping, the proliferation marker Ki-67 was introduced. However, there are well-known limitations in the use of this marker; these include inter-laboratory variability even among expert breast pathologists and a lack of consensus regarding the optimal cut-off point between positivity and negativity [9].

There is still some controversy over the most appropriate method to define ER and PR status and the reproducibility for the new diagnostic category of ‘HER-2 low’ BC is poor even among expert breast pathologists [10]. These obstacles have generated a search for novel avenues for BC subtyping. To overcome the IHC/ISH and cut-off issues, the proposal to use an mRNA-based assay has been suggested as a possible alternative in routine practice. The use of RT-qPCR, particularly in the form of the Xpert BC STRAT4 assay, has successfully been validated. This assay has been used in the Europe-wide EQA study and has shown correlation and reproducibility with IHC/ISH in several European laboratories; however, the number of cases assessed so far has been limited [11].

The recent development of the APIS Breast Cancer Subtyping Kit presents a realistic molecular alternative to Allred score for IHC [12] /ISH. It is an RNA-based diagnostic assay that assesses mRNA expression of the standard IHC biomarkers HER-2, ER and PR. In addition, Ki-67 plus another three genes provide further data used to generate a proliferation signature for every sample. This assessment can be performed on pre-operative core needle biopsies (CNB) or resected formalin-fixed paraffin-embedded (FFPE) breast tumour tissue. Laboratory personnel can perform the assessment of cancer samples using this kit; there is no requirement for the involvement of a specialist histopathologist.

To test APIS BC subtyping in clinical practice and eliminate selection bias, we decided to analyse 100 consecutive clinical cases. The total number was dictated by the number of APIS Kits available and the amount of available molecular staff overtime.

As a result, we present the first clinical study of 100 consecutive, prospective, pre-operative BC core biopsies analysed using the APIS mRNA assay. Samples were assessed in parallel with the routine IHC/ISH workflows, thus enabling patient discussion of all the data during the routine breast MDT meetings.

## 2. Results

### 2.1. APIS Molecular Characteristics

The run validities of the RNA results were available for 99 cases tested. One sample failed to produce sufficient RNA for analysis.

Using the BC Subtyping Kit analysis software (https://www.apisassay.com/breast-cancer-subtyping, accessed on 7 January 2024) developed by APIS Assay Technologies (Manchester, UK), the results calling produced 76 samples that were ER-positive (above −1.98; the cut-off value), 65 samples that were PR-positive (above −0.63) and 14 samples that were HER-2-positive (above 2.00). The proliferation signature (MKI67, PCNA, CCNA2 and KIF23) produced a ‘low’ category in 21 cases versus a high category in 78 cases. See Table 1.

Molecular classification according to this software produced the following subtypes: 39 samples Luminal A, 45 Luminal B (11 HER-2 + ve and 34 HER-2 − ve), 11 triple negative (TNBC) and 4 HER-2-enriched.

### 2.2. Pathological and IHC Characteristics

Results are shown in Table 2.

In this consecutive series of 100 cases, 3 BCs were Grade 1, 57 were Grade 2 and 38 were Grade 3. Two were nodal metastases. The large majority (89) were invasive carcinoma of no special type (formerly known as invasive ductal carcinomas (IDC)), 9 were invasive lobular carcinoma (ILC) and 2 were mucinous carcinoma.

The large majority of BCs were ER-positive (Allred score 7/8 in 8 cases and 8/8 in 78 cases), 14 cases were ER-negative. PR values were as follows: 0/8 in 14 cases, low (2-6/8) in 21 and high (7-8/8) in 75 cases [12,13].

HER-2 IHC was negative (score 0) in 59 cases, 1+ in 20 cases and 2+/ISH-negative in 5 cases while 15 cases were scored positive (3+). [14]

IHC Ki-67-proliferation was high (>20%) in 48 cases and low (<20%) in 52 cases.

### 2.3. Comparison of APIS mRNA Scores for ER, PR, HER-2, Ki-67 and IHC-Based Results

Each biomarker was recorded as either concordant or discordant when IHC and mRNA results were compared (See Table 2)

Briefly, there was 96% overall concordance between IHC and APIS across ER/PR and HER-2 assays: 97% ER, 89% PR and 100% HER-2.

Discordant Cases: 3/100 cases had results that were discordant between ER-IHC and ER-APIS. One sample was scored by IHC as luminal (ER score 7/8) but was ER-APIS-negative in both of two subsequent biopsies (case 6 and 71). This sample was also discordant for both biopsies in the PR assays. A further sample was also scored ER-IHC-positive (score 8/8) and ER-APIS-negative; case 60, a recurrent IDC in which most of the specimen was scar tissue with a very small amount of cancer cells. This sample was also discordant for PR (IHC-PR-positive, PR-APIS-negative). One sample (case 93, ILC with mixed features; both classical-ER+/HER-2 negative and pleomorphic ER-/HER-2+) was ER-IHC-positive but ER-APIS-negative.

There were an additional nine cases that were discordant PR-IHC versus PR-APIS. Further, 3% of cases were discordant between Ki-67 and APIS results when compared with MKI67 alone, but this rose to 24% discordance when the four-gene RNA proliferation signature was compared to Ki-67-IHC.

## 3. Discussion

The pandemic resulted in a huge backlog of patients awaiting surgery and subsequently a large number of specimens requiring histological evaluation; this has put an enormous strain on histopathology laboratories across the UK and, unfortunately, a consequence being that some cancers were taking longer to be diagnosed.

The APIS Breast Cancer Subtyping Kit provides data for all the biomarkers routinely used in breast cancer staging, allowing patient management based on robust and reproducible results. So far, those results have been provided through IHC and microscopic examination by highly trained histopathologists. The use of an RNA-based assay reduces the need for the involvement of this cohort of specialists, thereby releasing their time for other essential work.

Moreover, eliminating the subjective interpretation of IHC tests produces molecular results that can be used by healthcare professionals to tailor the treatment and management of each breast cancer patient according to their molecular signature.

Now that gene expression profiling is widely accepted and understood, there is an increasing demand from healthcare professionals to use the information from molecular tests to inform their treatment of patients or to enrol them in clinical trials. The APIS BC Subtyping Kit allows the provision of molecular intrinsic subtypes on a routine basis and therefore will go some way towards meeting this demand.

The data reported here indicate that there is an extremely high level of concordance in the results produced by two different methodologies, but it also identified a small number of clinically relevant discordances. While we are focused on the discordances with a view to the perfectibility of molecular testing, we should not let this take away from the validation provided by the remarkably high level of concordance.

In the evaluation of our discordant cases, we categorised them into clinically relevant discordances or discordances requiring improvements/refinement.

### 3.1. Clinically Relevant Discordances

Case 1 was assessed by histology as Grade 3 IDC with basal-like features being noted; it was recorded by IHC as the luminal subtype (ER score 7/8) with high proliferation. However, this sample was reported as a triple negative (basal-like) molecular subtype by the APIS RNA assessment on two biopsies from the same patient (cases 6 and 71). See Figure 1.

It is important to note that histologically luminal BCs classified as basal-like by expression analysis have been demonstrated to be those more responsive to chemotherapy, as shown by the GIADA neoadjuvant study [15]. This case is similar to one previously reported by Kim et al. [8] in which clinical follow-up showed worse overall survival than would be expected for a luminal subtype BC.

Case 2 was ER-IHC-positive but ER-APIS-negative. The patient had a previous hormone receptor-positive HER-2-negative IDC with a recurrence occurring in the scar tissue. A review of the slide showed most of the specimen was scar tissue with a limited amount of IDC and hence the possible cause of discordant results may be due to the low tumour cellularity of the sample. See Figure 2.

Case 3 was an ILC with mixed features: a small proportion of classical-ER+/HER-2-negative ILC cells and a larger proportion of pleomorphic, ER-negative/HER-2-positive cells. The APIS assay scored this ER- and PR-negative/HER-2-positive. Given that the latter component was more prevalent in the sample, the APIS-negative result is not surprising, i.e., the heterogeneity of the sample has to be taken into consideration when using and assessing samples with the APIS assay. See Figure 3. In this case, the hyper-cellular (ER-negative, HER-2-positive) was detected using APIS subtyping whilst the hypo-cellular (ER-positive) was not detected. Pathologists should therefore pay special attention to rare subtypes like pleomorphic ILC that can have both hyper- and hypo-cellularity.

Because low cellularity can affect molecular results, pathologists need to be aware that an in situ test such as IHC is preferable in that case.

### 3.2. Discordances Requiring Improvements/Refinement

When discussing those areas of discordance that we defined as ‘discordances requiring improvements/refinement‘, this mainly involves the APIS PR results compared to IHC and the Ki-67 IHC analysis compared to the results of the proliferation signature; these represent the majority of the discordant cases in this study (10/100 and 24/100, respectively).

Of the nine PR discordant cases, seven had weak and moderate nuclear staining, producing a score between two and six on IHC, but they were all scored negative by the APIS assay. It has been widely noted that the behaviour of BCs with low levels of hormone receptors is more consistent with features of triple negative breast carcinomas rather than with luminal carcinomas. Even the ASCO/CAP recommendations suggest including a specific comment when encountering low levels of hormone receptors [13,14].

Currently, the discrepancy between the final two cases is unexplained, although one had very few tumour cells in the samples and this may have contributed to the negative PR result in the APIS assay.

The limitations of measuring proliferation markers Ki-67 are well known. They include inter-laboratory variability even among expert breast pathologists and there is a lack of consensus around Ki-67 values which represent optimal cut-off points for positivity and negativity [9]. A recent study recommended the use of PHH3 antibody to mitotic count accuracy [16]

The assessment of the proliferation signature (MK67, PCNA, CCNA2 and KIF23) is most likely the reason for the larger number of discrepant results, as these markers take into account genes involved in aspects of the cell cycle other than that of Ki-67, i.e., PCNA [17], Kinesin 23 (KIF23) [18] and Cyclin A2 (CCNA2) [19].

It is accepted that a gene signature better reflects the proliferative nature of a sample than just a single marker (Ki-67) [16,20].

Our study considered the challenges implementing the APIS assay into clinical practice, taking into account, particularly, the time-effectiveness of the assay; the assay could produce data in a similar time-frame and similar costs to that produced by IHC. The introduction of the APIS BC Subtyping Kit into routine analysis in place of IHC could potentially save pathologists’ time and alleviate pressure on IHC pathology laboratories which have been significantly impacted by post-COVID backlog cases and led to delays in patient management. The application on core biopsies also has potential implications on the use of this assay in the context of the best selection of neoadjuvant treatment in luminal and TNB carcinomas.

Nevertheless, the assay does have limitations, particularly around the morphological heterogeneity and tumour cellularity of samples. There are clinical implications such as a patient might be treated as having non-luminal BC and therefore be deprived of the benefit of hormone therapy if a heterogeneous tumour was assessed incorrectly as in Case 2 in our study. In situ analysis is still superior to in vitro nucleic acid-based analysis in this respect, i.e., when the specimen is scarce in cellularity or morphologically heterogeneous.

It is possible that IHC would still have to be used in cases where the histology indicates a heterogeneous sample or one with low tumour burden.

Moreover, the cohort of BCs analysed in this study were mostly Grade 2 and Grade 3 BC; hence, the impact of this mRNA assessment in the wider cohort of BC remains to be determined.

The assay currently only provides a binary (positive–negative) mRNA result. It has been recommended to the manufacturers that they design a report where the level of ER, PR and HER-2 is graded in a way that mirrors the IHC-based reports. In other words, where ER and PR reported with a score from 2 to 8, HER-2 reported with low scores (1+, 2+/ISH) and positive 2 + ISH + and 3+, thereby truly offering a like-for-like replacement to IHC/ISH assessment.

The usability of the APIS tests is clearly impacted by the cost/benefit position. As such, APIS have committed to providing their assay to the NHS on a cost neutral basis.

## 4. Material and Methods

### 4.1. Samples

A total of 98 samples of FFPE pre-operative breast core biopsies and 2 core biopsies of lymph nodes were assessed histologically and the presence and grade of cancer confirmed. Current routine processes assessed all the samples using IHC/ISH and they were simultaneously assessed using the APIS mRNA-based assay.

The mRNA data generated by the APIS assay were included in the patient pathological report, and clinical discussion of the data occurred. It is important to note that national guidelines were followed and only the IHC/ISH results were employed in relation to patient management.

### 4.2. Immunohistochemistry and ‘In Situ‘ Hybridisation

Immunohistochemistry for HER-2, ER, PR and Mib-1 (Ki-67) was performed using an automated staining module as per standard protocols for routine assessment of these samples. Nuclear staining for ER and PR was scored using the Allred scoring system for BC based on the assessment of the intensity of nuclear staining and the proportion of immuno-stained cells [11].

ER IHC was divided into negative (0/8) and strongly positive (7-8/8) groups; no cases of weak or intermediate nuclear staining were recorded. PR IHC was divided into three groups based on the staining seen: negative (0/8), weak-moderate (2-6/8) and strong (7-8/8).

Ki-67 IHC was divided into two groups: low (<20%) and high (>20%). The threshold for positivity of >1% was adopted as described previously regarding ASCO/CAP ER/PR [9].

For HER-2, the standard ASCO⁄CAP scoring system was applied: 0, 1+, 2+ and 3+.

ISH was used to further assess BCs exhibiting HER-2 2 + IHC.

All IHC slides from each BC case underwent histological evaluation by an experienced BC pathologist.

### 4.3. RNA Extraction

Total RNA was isolated from FFPE tissue sections measuring 10 µm in thickness, ensuring that an area of the specimen with tumour content of ≥20% was analysed. The automated extraction of RNA was performed using the Maxwell48 RSC RNA FFPE Kit following the manufacturer’s recommended protocol. Subsequently, the RNA content within each eluate was quantified and normalised to a concentration of 2.5 ng/µL. Samples were stored at −80 °C until required.

### 4.4. Gene Expression by RT-qPCR

The quantification of mRNA expression levels was performed using the APIS Breast Cancer Subtyping Kit. This analysis uses RT-PCR for the measurement of *ESR1*, *PGR*, *ERBB2* and *MKI67*, as well as three additional targets found within the proliferation signature (*CCNA2*, *KIF23* and *PCNA*), along with two reference genes (*IPO8* and *PUM1*).

The assay was set up as per the manufacturer’s protocol and the RT-qPCR reaction was analysed using the QuantStudio™ 5 Dx real-time PCR system (QS5™Dx; Thermo Fisher Scientific, Waltham, MA, USA).

The protocol requires each specimen to undergo duplicate analysis for each gene assay. A maximum of 10 patient samples and a positive and negative control could be analysed at any one time.

APIS proprietary software (version 1.2.1) was used to determine levels of gene expression for each sample, utilising the reference genes to generate a ΔCt. Briefly, this is achieved through the subtraction of the average cycle threshold (Ct) value of the duplicate measurements of the target of interest from the mean Ct value obtained from duplicate measurements of the reference genes. Binary target calls (positive/negative) were reported based on previously established ΔCt cut-off values specific to each target gene.

A logistic model using ΔCt values for *MKI67*, *CCNA2*, *PCNA* and *KIF23* was employed to calculate a proliferation score within the range of 0 to 1. 

A score below 0.5 indicates low proliferation, while a score above 0.5 indicates high proliferation. The molecular subtype classification was determined by considering the combined statuses of ESR1, PGR, ERBB2 and MKI67.

## Figures and Tables

**Figure 1 ijms-25-02616-f001:**
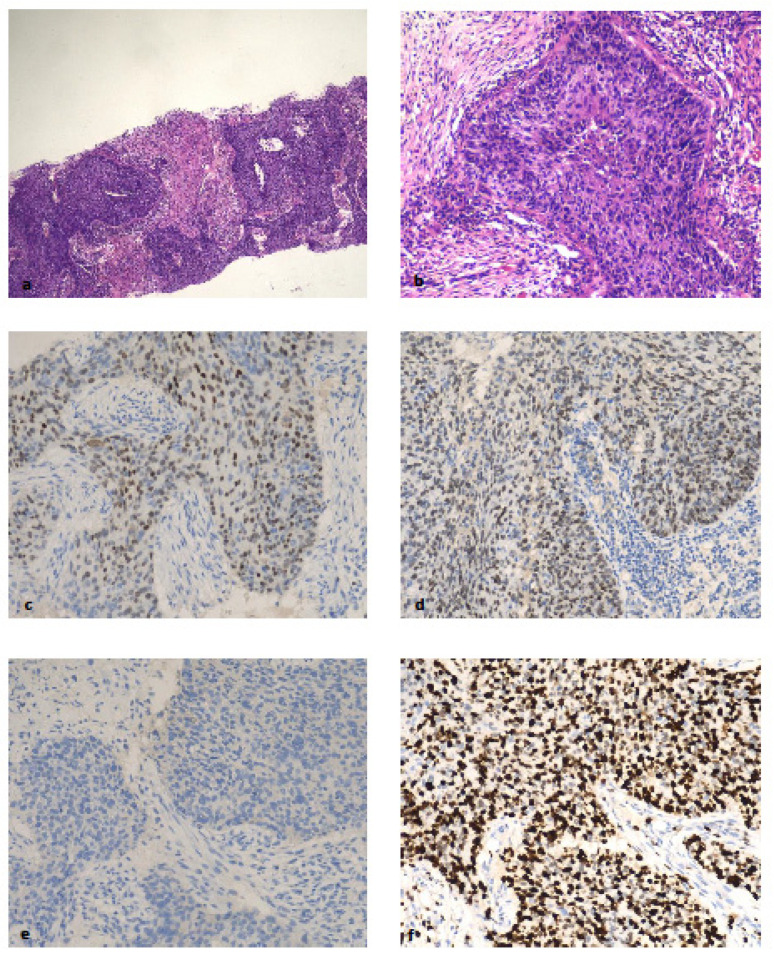
Case 1 discordant case between IHC (luminal ER and PR-positive) and APIS TNBC. (**a**) Breast core biopsy showing an invasive carcinoma Grade 3 with basaloid architecture (H&E 10×); (**b**) tumour cells show palisaded architecture at the periphery of tumour islands; (**c**) estrogen receptors detected nuclear staining in >75% with a moderate nuclear intensity of staining; (**d**) progesterone receptor >75% of cells show positive and moderate staining; (**e**) there is no HER-2 protein overexpression (HER-2-negative); (**f**) Ki-67 is noted in almost 100% of nuclei of the tumour cells. (**b**–**f**): 400×.

**Figure 2 ijms-25-02616-f002:**
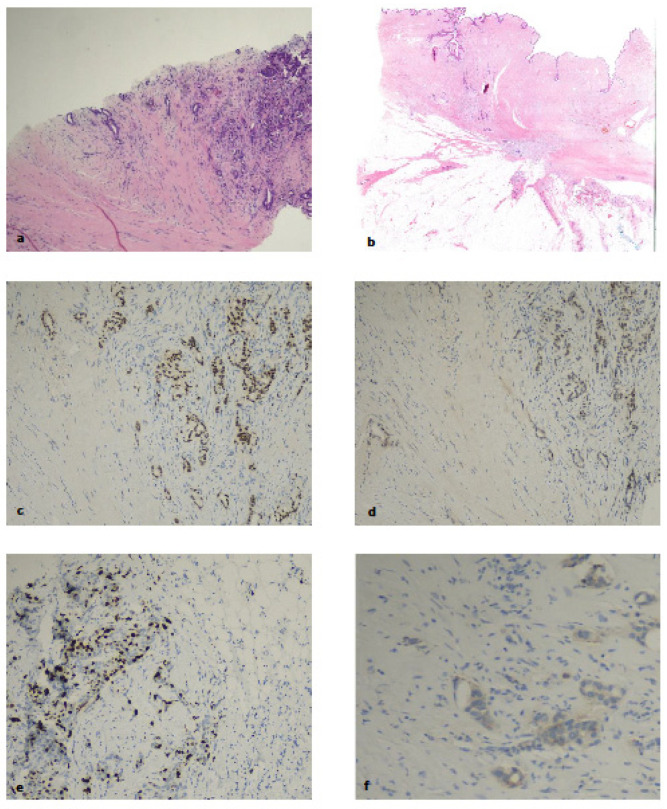
Case 2 discordant IHC ER and PR-positive and APIS ER/PR-negative. Core biopsy of recurrent breast carcinoma. (**a**) Prominent scar tissue on the left and tumour tissue on the right (H&E 10×); (**b**) recurrence is seen in the cutaneous scar and subcutaneous fat of the surgical specimen (H&E whole mounted section); (**c**) oestrogen receptor-positive; (**d**) progesterone-positive; (**e**) Ki-67 up to 20% of hotspot areas; (**f**) HER-2 protein low weak cytoplasmic membrane staining (score 1+). (**c**–**f**) 400×.

**Figure 3 ijms-25-02616-f003:**
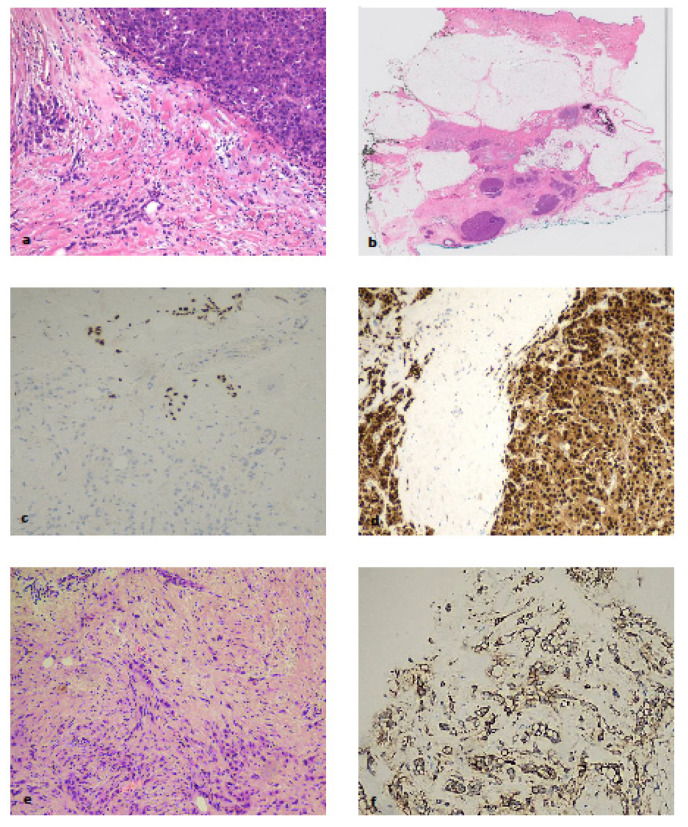
Discordant APIS/IHC case for the oestrogen receptor. (**a**) Core biopsy of a pleomorphic invasive lobular carcinoma, with solid and diffuse tumour architecture (H&E 20×); (**b**) subsequent surgical specimen confirms nodular and diffuse architecture (H&E whole mounted section); (**c**) IHC for oestrogen receptor with no (bottom) and strong (top) nuclear staining; (**d**) androgen receptor; (**e**) discrete infiltration of tumour cells; (**f**) HER-2 protein strong and complete circumferential cytoplasmic membrane staining (score 3+). (**c**–**f**) 400×.

**Table 1 ijms-25-02616-t001:** ΔCt cut-off values for the APIS BC Subtyping Kit targets.

Target	Positive/Negative Cut-Off Values
ESR1	−1.98
PGR	−0.63
ERBB2	2.00
MKI67	−0.64

**Table 2 ijms-25-02616-t002:** Main immunohistochemical and mRNA features.

Immunohistochemistry						APIS Breast Cancer Subtyping Kit Analysis					
Serial No	Type/Met	ER	PR	HER-2	MIB1 IHC	ESR1	PGR	ERBB2	MK167	Proliferation	Molecula Subtype
1	IDC G2	8/8	8/8	low 2 + ISH-neg	low	+ve	+ve	−ve	low	low	Luminal A
2	IDC G2	8/8	8/8	low 2 + ISH − ve	low	+ve	+ve	−ve	high	high	Luminal B HER-2 − ve
3	IDC G3	8/8	8/8	low 1+	high	+ve	+ve	−ve	high	high	Luminal B HER-2 − ve
4	IDC G3	0/8	6/8	−ve	high	−ve	−ve	−ve	high	high	TNBC
5	IDC G3	8/8	8/8	−ve	high	+ve	+ve	−ve	high	high	Luminal B HER-2 − ve
6	IDC G3	7/8	7/8	−ve	high	−ve	−ve	−ve	high	high	TNBC
7	IDC G2	8/8	8/8	low 1+	high	+ve	+ve	−ve	high	high	Luminal B HER-2 − ve
8	IDC G3	8/8	8/8	low 1+	high	+ve	+ve	−ve	high	high	Luminal B HER-2 − ve
9	IDC G2	8/8	0/8	−ve	low	+ve	−ve	−ve	low	high	Luminal A
10	IDC G2	8/8	0/8	low 1+	low	+ve	−ve	−ve	low	low	Luminal A
11	IDC G3	0/8	0/8	+ve 3+	low	−ve	−ve	+ve	low	high	HER-2-enriched
12	IDC G2	8/8	0/8	−ve	low	+ve	−ve	−ve	low	high	Luminal A
13	IDC G2	8/8	0/8	−ve	low	NA	NA	NA	NA	NA	NA
14	IDC G2	8/8	4/8	−ve	high	+ve	−ve	−ve	high	high	Luminal B HER-2 − ve
15	IDC G1	7/8	8/8	−ve	low	+ve	+ve	−ve	low	low	Luminal A
16	IDC G2	8/8	8/8	−ve	low	+ve	+ve	−ve	low	low	Luminal A
17	IDC G3	0/8	0/8	−ve	high	−ve	−ve	−ve	high	high	Triple negative
18	IDC G3	8/8	7/8	−ve	high	+ve	+ve	−ve	high	high	Luminal B HER-2 − ve
19	IDC G2	8/8	8/8	−ve	high	+ve	+ve	−ve	high	high	Luminal B
20	IDC G3	8/8	8/8	−ve	high	+ve	+ve	−ve	high	high	Luminal B HER-2 − ve
21	IDC G2	8/8	8/8	−ve	low	+ve	+ve	−ve	low	low	Luminal A
22	IDC G2	8/8	0/8	−ve	low	+ve	−ve	−ve	low	high	Luminal A
23	IDC G2	8/8	8/8	−ve	low	+ve	+ve	−ve	low	high	Luminal A
24	IDC G3	8/8	4/8	−ve	high	+ve	−ve	−ve	high	high	Luminal B HER-2 − ve
25	IDC G2	8/8	6/8	low 1+	low	−ve	+ve	−ve	low	high	Luminal A
26	IDC G2	8/8	8/8	−ve	low	+ve	+ve	−ve	low	high	Luminal A
27	IDC G3	8/8	8/8	−ve	low	+ve	+ve	−ve	low	low	Luminal A
28	IDC G2	8/8	8/8	−ve	low	+ve	+ve	−ve	low	high	Luminal A
29	IDC G3	0/8	0/8	+ve 3+	low	−ve	−ve	−ve	low	high	HER-2-enriched
30	IDC G2	8/8	7/8	−ve	low	+ve	+ve	−ve	low	low	Luminal A
31	IDC G3	8/8	8/8	+ve 3+	high	+ve	+ve	+ve	high	high	Luminal B HER-2 + ve
32	IDC G3	0/8	2/8	−ve	high	−ve	−ve	−ve	low	high	TNBC
33	IDC G2	8/8	8/8	−ve	high	+ve	+ve	−ve	high	high	Luminal B HER-2 − ve
34	IDC G2	8/8	8/8	low 2 + ISH −ve	high	+ve	+ve	−ve	high	high	Luminal B HER-2 − ve
35	IDC G2	8/8	0/8	−ve	low	+ve	−ve	−ve	low	high	Luminal A
36	IDC G3	8/8	8/8	−ve	high	+ve	+ve	−ve	high	high	Luminal B HER-2 − ve
37	IDC G2	0/8	0/8	−ve	low	−ve	−ve	−ve	low	low	Triple negative
38	IDC G2	7/8	8/8	low 1+	low	+ve	+ve	neg	low	low	Luminal A
39	IDC G3	7/8	8/8	low 1+	high	positive	positive	neg	high	high	Luminal B HER-2 − ve
40	IDC G3	0/8	0/8	+ve 3+	high	−ve		+ve	high	high	Luminal B HER-2 + ve
41	IDC G2	8/8	8/8	−ve	high	+ve	+ve	−ve	high	high	Luminal B HER-2 −ve
42	IDC G3	8/8	8/8	+ve 3+	high	+ve	+ve	+ve	high	high	Luminal B HER-2 + ve
43	IDC G3	0/8	3/8	low 1+	high	−ve	−ve	−ve	high	high	TNBC
44	IDC G2	8/8	6/8	2 + ISH −ve	low	+ve	−ve	−ve	low	low	Luminal A
45	ILC G2	8/8	8/8	−ve	low	+ve	+ve	negative	low	high	Luminal A
46	ILC G2	8/8	8/8	−ve	low	+ve	+ve	−ve	low	high	Luminal A
47	IDC G2	8/8	8/8	low +1	high	+ve	+ve	−ve	high	high	Luminal B HER-2 − ve
48	IDC G3	0/8	3/8	−ve	high	−ve	−ve	−ve	high	high	TNBC
49	ILC G2	8/8	8/8	low 1+	low	+ve	+ve	−ve	low	low	Luminal A
50	IDC G3	8/8	8/8	−ve	high	+ve	+ve	−ve	high	high	Luminal B HER-2 − ve
51	IDC G3	8/8	6/8	+ve 3+	high	+ve	−ve	+ve	high	high	Luminal B HER-2 + ve
52	IDC G3	8/8	8/8	low 1+	high	+ve	+ve	−ve	high	high	Luminal B HER-2 − ve
53	IDC G3	0/8	2/8	+ve 3+	high	−ve	+ve	+ve	high	high	Luminal B HER-2 + ve
54	IDC G2	8/8	8/8	−ve	high	+ve	+ve	−ve	high	high	Luminal B HER-2 − ve
55	IDC G3	8/8	8/8	low 1+	high	+ve	+ve	−ve	high	high	Luminal B HER-2 − ve
56	IDC G2	8/8	8/8	−ve	high	+ve	+ve	−ve	high	high	Luminal B HER-2 − ve
57	IDC G2	0/8	3/8	+ve 3+	high	−ve	+ve	+ve	high	high	Luminal B HER-2 + ve
58	IDC G3	8/8	8/8	+ve 3+	high	+ve	+ve	+ve	high	high	Luminal B HER-2 + ve
59	Mucinous ca G2	8/8	8/8	Low 1+	low	+ve	+ve	−ve	low	high	Luminal A
60	IDC G2	8/8	8/8	low 2 +, ISH −ve	low	−ve	−ve	−ve	low	high	Triple negative
61	IDC G2	8/8	8/8	+ve 3+	low	+ve	+ve	+ve	low	high	Luminal B HER-2 + ve
62	IDC G2	8/8	8/8	low 1+	low	+ve	+ve	−ve	low	low	Luminal A
63	IDC G2	8/8	8/8	−ve	low	+ve	+ve	−ve	low	high	Luminal A
64	IDC G3	8/8	8/8	−ve	high	+ve	+ve	−ve	high	high	Luminal B
65	IDC G2	8/8	2/8	−ve	low	+ve	−ve	−ve	low	high	Luminal A
66	IDC G1	8/8	5/8	−ve	low	+ve	+ve	−ve	low	low	Luminal A
67	IDC G3	8/8	8/8	−ve	high	+ve	+ve	−ve	high	high	Luminal B HER-2 − ve
68	IDC G2	8/8	8/8	low 1+	high	+ve	+ve	−ve	high	high	Luminal B HER-2 − ve
69	IDC G3	0/8	0/8	−ve	high	−ve	−ve	−ve	high	high	TNBC
70	ILC G2	8/8	8/8	low 1+	low	+ve	+ve	−ve	low	low	Luminal A
71	IDC G3	7/8	7/8	−ve	high	−ve	−ve	−ve	high	high	TNBC
72	ILC G2	8/8	8/8	−ve	low	+ve	+ve	−ve	low	high	Luminal A
73	ILC G2	8/8	8/8	−ve	low	+ve	+ve	−ve	low	high	Luminal B HER-2 − ve
74	Nodal met IDC G2	8/8	8/8	−ve	high	+ve	+ve	−ve	high	high	Luminal B HER-2 − ve
75	IDC G3	8/8	5/8	low 1+	high	+ve	+ve	−ve	high	high	Luminal B HER-2 − ve
76	IDC G3	8/8	3/8	−ve	high	+ve	−ve	−ve	high	high	Luminal B HER-2 − ve
77	IDC G2	8/8	8/8	−ve	low	+ve	+ve	−ve	low	high	Luminal A
78	ILC G2	8/8	7/8	−ve	low	+ve	+ve	−ve	low	high	Luminal A
79	IDC G2	8/8	8/8	−ve	low	+ve	+ve	−ve	low	low	Luminal A
80	IDC G2	8/8	3/8	−ve	low	+ve	+ve	−ve	low	low	Luminal A
81	ILC G2	8/8	3/8	−ve	low	+ve	−ve	−ve	low	high	Luminal A
82	IDC G3	7/8	7/8	−ve	high	+ve	+ve	+ve	high	high	Luminal B HER-2 − ve
83	IDC G3	8/8	8/8	+ve 3+	high	+ve	+ve	+ve	high	high	Luminal B HER-2 + ve
84	IDC G2	8/8	0/8	−ve	low	+ve	−ve	−ve	low	low	Luminal A
85	IDC G2	8/8	8/8	low 1+	low	−ve	+ve	−ve	low	high	Luminal A
86	IDC G3	8/8	8/8	−ve	low	+ve	+ve	−ve	low	high	Luminal A
87	IDC N met	0/8	0/8	2 + ISH −ve	high	−ve	−ve	−ve	high	high	TNBC
88	IDC G1	8/8	4/8	−ve	low	+ve	+ve	−ve	low	low	Luminal A
89	IDC G3	8/8	5/8	+ve 3+	high	+ve	−ve	+ve	high	high	Luminal B HER-2 + ve
90	IDC G2	8/8	8/8	−ve	high	+ve	+ve	−ve	high	high	Luminal B HER-2 − ve
91	Muc G2	8/8	8/8	−ve	low	+ve	+ve	−ve	low	low	Luminal A
92	IDC G2	8/8	8/8	−ve	high	+ve	+ve	−ve	high	high	Luminal B HER-2 − ve
93	Pleom ILC	8/8	3/8	+ve 3+	low	−ve	−ve	+ve	low	high	HER-2-enriched
94	IDC G3	7/8	6/8	low 1+	high	+ve	+ve	−ve	high	high	Luminal B HER-2 − ve
95	IDC G3	7/8	8/8	low 1+	high	+ve	+ve	−ve	high	high	Luminal B HER-2 − ve
96	IDC G2	8/8	4/8	−ve	low	+ve	−ve	−ve	high	high	Luminal B HER-2 −ve
97	IDC G2	8/8	8/8	−ve	high	+ve	+ve	−ve	high	high	Luminal B HER-2 − ve
98	IDC G2	8/8	8/8	low 1+	low	+ve	+ve	−ve	low	high	Luminal A
99	IDC G3	0/8	3/8	−ve	high	−ve	−ve	+ve	high	high	HER-2-enriched
100	ILC G2	8/8	8/8	−ve	low	+ve	+ve	neg	low	low	Luminal A

## Data Availability

Data is contained within the article.

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
