# Peer review of "Breast Cancer Molecular Subtyping in Practice: A Real-World Study of the APIS Breast Cancer Subtyping Assay in a Consecutive Series of Breast Core Biopsies"

_ijms, 2024, doi:10.3390/ijms25052616_

Round 1

Reviewer 1 Report

Comments and Suggestions for Authors

To the best of this reviewer's knowledge this study represents the first report on the comparison of standard IHC for biomarkers in breast cancer and the APIS assay, which is an RNA-based assay that reports molecular subtyping of breast cancer. 

The study us therefore novel in this respect and provides data on the concordance of the two assays.

I would suggest:

1) to move the figures from the discussion section to the result section

2) to stress more that careful selection should be performed of the areas of interest from which the tissue that has to be analyzed for an in vitro nucleic acid-based assay : it looks like that two cases has discordances most likely due to the low cellularity of the sample. Perhaps a message that could be rendered in this scenario is that whenever low tumor cellularity is detected an in situ assay such as IHC is always preferable.

3) to add a comment about costs and sustainability of such an assay in clinical practice, especially for this scenarios in which the author would see a contribution by this test.

Comments on the Quality of English Language

Some typos could be deleted, please double check carefully the whole manuscript.

Reviewer 2 Report

Comments and Suggestions for Authors

January 18, 2024

Ms. Ref. No.: ijms-2838649

Journal: International Journal of Molecular Sciences.

Title: Breast cancer molecular subtyping in practice: a real-world study of the APIS Breast Cancer Subtyping assay in a consecutive series of breast core biopsies.

Comments:

I appreciate the time and effort you put into writing this article on an important subject. I have a few observations regarding its content that I believe could enhance its quality. Please find them outlined in the following paragraphs.

1- tharticle  mentioned the three biomarkers (estrogen receptor (ER), progesterone receptor (PR), and HER2), while after that introducing four genes that create a novel proliferation signature, MKI67, PCNA, CCNA2, KIF23, why did researchers choose these biomolecules?

2-      Please reintroduce the association of 7 parameters (including estrogen receptor (ER), progesterone receptor (PR), and HER2 , MKI67, PCNA, CCNA2, KIF23) with breast cancer.

3-      With a sample size of 98 patients, it's important to understand how this number was determined. Can you please provide more information about the calculation process?

4-      The first paragraph of the "2.3. RNA Extraction" section is in a larger font size than the rest of the text. It would be better to use the same font size for consistency.

5-      Upon reviewing Table 1, we have come to the understanding that ERBB2 displays a positive value, while the rest of the subjects exhibit negative values. In light of this, we kindly request your input on how we can reintroduce this subject. We believe that with your valuable insights, we can explore effective strategies to ensure that this subject receives the attention it .

6-      To enhance the level of comprehensibility in the introduction, it is advisable to consider incorporating some of the following references:

·         https://doi.org/10.3390/ijms25021242

·         https://doi.org/10.1007/s12013-023-01171-y

·         https://doi.org/10.3390/ijms25021110
